# Pilot Implementation of a Nutrition-Focused Community-Health-Worker Intervention among Formerly Chronically Homeless Adults in Permanent Supportive Housing

**DOI:** 10.3390/ijerph21010108

**Published:** 2024-01-18

**Authors:** Jane E. Hamilton, Diana C. Guevara, Sara F. Steinfeld, Raina Jose, Farrah Hmaidan, Sarah Simmons, Calvin W. Wong, Clara Smith, Eva Thibaudeau-Graczyk, Shreela V. Sharma

**Affiliations:** 1McGovern Medical School, The University of Texas Health Science Center Houston, Houston, TX 77030, USA; diana.guevara@uth.tmc.edu (D.C.G.);; 2Bread of Life, Inc., Houston, TX 77002, USAfarrahhmaidan@yahoo.com (F.H.);; 3Temenos, CDC, Houston, TX 77003, USA

**Keywords:** food insecurity, homelessness, permanent supportive housing, program planning/implementation to address social determinants of health

## Abstract

Food insecurity is a known health equity threat for formerly chronically homeless populations even after they transition into permanent housing. This project utilized a human-centered design methodology to plan and implement a nutrition-focused community-health-worker (CHW) intervention in permanent supportive housing (PSH). The project aimed to increase access to healthy foods, improve nutritional literacy, healthy cooking/eating practices, and build community/social connectedness among 140 PSH residents. Validated food-security screening conducted by CHWs identified low or very low food security among 64% of 83 residents who completed the baseline survey, which is similar to rates found in a previous study among formerly homeless populations placed in PSH. Major themes identified through an analysis of resident feedback include (1) lack of needed kitchenware/appliances for food preparation, (2) knowledge gaps on how to purchase and prepare healthier food, (3) positive perceptions of healthy food options, (4) expanded preferences for healthy, easy-to-prepare foods, (5) regaining cooking skills lost during homelessness, (6) positive experiences participating in group activities, (7) community re-entry, and (8) resident ownership. Preliminary findings suggest the use of a human-centered design methodology for planning and implementing this multi-level CHW intervention helped reduce food insecurity, engaged participants in learning and adopting healthy and safe cooking and eating practices, and fostered social connectedness and feelings of community among formerly chronically homeless PSH residents.

## 1. Introduction

Food security, access to nutritious, safe, and culturally adequate food, is a critical social determinant of health [1]. Research supports a direct relationship between food security, diet quality, and physical and mental health [2,3,4], as well as social well-being [5]. However, food insecurity remains an intractable public health problem in the U.S. [6]. The U.S. Department of Agriculture (USDA) defines food insecurity as “a household-level economic and social condition of limited or uncertain access to adequate food” [7]. Food insecurity leads to disordered eating patterns and higher consumption of energy dense, nutrient-deficient foods, resulting in poor-health outcomes [8]. Further, a nutrient-poor diet among people experiencing food insecurity contributes to the development, and impedes the self-management of chronic illnesses including heart disease, hypertension, diabetes, chronic kidney disease, and depression [9,10,11,12]. 

People experiencing homelessness are at an increased risk for food insecurity, with the risk often persisting after this population transitions to stable housing [13]. Chronic homelessness describes people experiencing at least one year of continuous homelessness—or repeated episodes—while struggling with a disabling condition and often facing long-term food insecurity [14]. Even after they are housed, the formerly chronically homeless population may face continued food insecurity and nutritional deficiency due to functional impairments, limited life skills, and/or a lack of economic resources [13,15].

Permanent supportive housing (PSH) promotes housing stability by combining subsidized housing and flexible supportive services for people with histories of chronic homelessness [16,17]. However, high levels of food insecurity and poor diet quality among PSH residents are posited to partially explain the lack of definitive health gains for this population [13,18]. Prior research on the impact of PSH on food insecurity has demonstrated low or very low food security among 67% of formerly chronically homeless adults (ages 45+ years) in two PSH programs in Los Angeles, California (*n* = 237), exceeding rates among similarly aged low-income adults in the general population and currently homeless adults [13]. Research also found very low food insecurity among 39 youth in PSH (ages 18–25) in San Francisco, California [19]. 

Increased morbidity and mortality among the chronically homeless and recently housed populations [17,20,21,22,23,24,25] points to a need for innovative strategies to address the ongoing health equity threat from food insecurity. Using a human-centered design, we planned and piloted a nutrition-focused community-health-worker (CHW) intervention in PSH tailored to the lived realities, needs, and preferences of PSH residents with histories of chronic homelessness [26]. We are not aware of other food insecurity interventions to date implemented in PSH serving formerly chronically homeless adults. 

## 2. Methods

### 2.1. Setting

This pilot project was conducted from May 2022 to April 2023 at a PSH program (comprising two multi-story apartment buildings with 140 rooms total) in Houston, Texas. The PSH program is a service provider partner of the regional U.S. Housing and Urban Development (HUD) Continuum of Care (CoC), The Way Home [27]. The PSH program adheres to a Housing First model by providing immediate access to permanent, subsidized, independent housing with no prerequisites such as mandating treatment participation or requiring sobriety [28]. This project was prioritized for community benefits funding from the Houston Methodist Hospital system because the residents comprise a socially vulnerable population and the zip code location of the PSH apartment buildings are within a “Food Desert,” due to large proportions of households within this zip code having low incomes, inadequate access to transportation, and a limited number of food retailers providing fresh produce and healthy groceries for affordable prices [29].

### 2.2. Participants

All 140 residents at one of the two PSH buildings in which the project was being implemented were eligible for project participation. There were no criteria for excluding participants. The majority of participants were recruited during the first and second quarters of program implementation by CHWs through door-to-door outreach, through flyers in the PSH building and at PSH community events. During program enrollment, informed consent was obtained by the CHWs as part of a food security survey and needs-assessment process and was documented for each client. All 140 PSH residents targeted for participation had a history of chronic homelessness and were prioritized for PSH through the regional HUD CoC using a validated prioritization tool [27]. A person is considered chronically homeless if they have a disabling condition and have been continuously homeless for more than a year or have had at least four episodes of homelessness over 3 years [14]. Because people experiencing chronic homelessness often have poor health status, poor quality of life, and premature mortality [30], food insecurity must be addressed to improve health outcomes for this vulnerable population. Demographics collected at the PSH program at the initiation of the project found that 78% of the 140 PSH residents had a serious mental illness (SMI); 58% had a substance use problem; and 75% had at least one chronic condition. Many residents were of advanced age; 29% were 62+ years; 37% were 55–61 years; and 15% were 45–54 years. Additionally, PSH residents had a high mortality risk (15 residents had died in the previous two years). The diverse resident population was 78% male, 20% female, 2% transgender, 77% Black/African American, 16% White, 2% Asian, and 1% Native American (96% Non-Hispanic). At the time of program implementation, only half of PSH residents in the two buildings where the pilot program was being implemented were connected with nutritional benefits through the Texas Supplemental Nutrition Assistance Program (SNAP). This pilot project was implemented as part of the larger recovery-oriented integrated behavioral health program, the UTHealth Houston-Homeless Outpatient Mental Health Expansion Services (UTHealth Houston HOMES) program, which is funded by the Substance Abuse and Mental Health Services Administration (SAMHSA; Grant IDs: SM080721; SM088614). The project was reviewed by the UTHealth IRB (HSC-MS-18-0751) and exempted as non-human research. 

### 2.3. Human-Centered Design

Program activities were implemented from July 2022 to April 2023 with 140 PSH residents following hiring and initial training using a human-centered design methodology to ensure the integration of evidence-based practices and participant feedback into all project activities [26,31]. This implementation strategy provided a creative approach for addressing food insecurity among PSH residents, providing a way for the project funder, university-based project leads, community-based agency partners, community health workers, and PSH residents to all come together to prioritize goals and project activities given the constraints of the low-resource environment in which the project was being implemented [31]. According to *The Field Guide to Human-Centered Design* (2015), project implementation occurs across three main phases: *Inspiration, Ideation*, and *Implementation* [32]. During the *Inspiration* phase, university and community-based partners came together with the goals of reducing food insecurity and increasing healthy cooking/eating among PSH residents. It was during this phase that the community-benefits grant was identified and the solutions for addressing food insecurity among PSH residents were discussed with multiple stakeholders. Nutrition faculty at the school of public health provided guidance on how to narrow the scope of the project to maximize impact and to build on the best practices implemented with vulnerable populations [32]. During the *Ideation* phase, a project plan was created during a series of stakeholder meetings to guide implementation activities across the four quarters of the 12-month project [32] (see Project Timeline Figure 1). During the *Implementation* phase, systematic feedback was obtained from PSH residents through a formal needs’ assessment and comment cards collected during project activities and by the project team during food- and kitchen-item-distribution events. PSH-resident feedback was reviewed during weekly planning meetings with project leaders and CHWs [32]. The systematic collection and review of PSH-resident feedback to improve project activities was at the core of the human-centered design process used to ensure the adoption of healthy cooking and eating practices. Additionally, all planning and implementation activities occurred at the PSH buildings and adjacent community-based non-profit in an effort to best understand the challenges faced by the PSH program administrators and PSH residents.

### 2.4. Pilot Intervention 

Three CHWs delivered a multi-level, pilot, nutrition-focused, educational-empowerment intervention to reduce barriers to health equity by increasing access to healthy foods, improving nutritional literacy and healthy cooking/eating practices, and building community/social connectedness through group activities. Intervention activities included the training and development of CHWs provided by university-based partners on survey administration, *Cooking Matters for Adults* SNAP-Education (SNAP-Ed) curriculum [33], resident engagement, and nutrition education. Interventions targeting PSH residents were divided into individual-level and group-level activities to ensure residents were connected with the necessary food and kitchen items and gained knowledge to improve safe/healthy cooking and eating practices (see project logic model Figure 2). Individual-level nutrition activities were designed to be delivered to PSH residents in their single-room-occupancy (SRO) apartments including pantry and cooking safety inspections, meal planning, nutrition-label reading, stocking pantries with healthy foods, cooking instructions, and linkage to individual “ask a dietitian” sessions. CHWs linked PSH residents with healthy food, necessary kitchen items (pots, pans, dishes, flatware, and small kitchen appliances) and nutrition benefits (SNAP and the Supplemental Nutrition Program for Women, Infants, and Children [WIC]). Group-level activities included cooking classes, weekly nutrition-education sessions, community meals, and sampling of nutritious and culturally inclusive foods obtained from the regional food bank, grocery store pop-ups, and field trips, creation of a resident cookbook, and community gardening.

### 2.5. Community-Health-Worker Training 

During the first quarter of the project, a 3-day live, virtual workshop, *Cooking Matters for Adults*, developed from the SNAP-Ed-curriculum, was delivered by the University of Minnesota Extension to three CHWs recruited to work with the PSH residents. The workshop topics included: culinary nutrition skills, building recipes, and basic nutrition education, and shopping for nutritious foods on a limited budget to stretch SNAP funds [33]. CHWs also completed virtual, evidence-based online-training nutrition modules based on a culinary-medicine approach [34] developed by the Nourish Program at the Michael & Susan Dell Center for Healthy Living at The University of Texas Health Science Center at Houston (UTHealth Houston) School of Public Health [35]. Prior to implementing community gardening activities, the dietitian engaged CHWs in an interactive tour of a university-based community garden used for public health nutrition education. In-person training was provided by the project director on best practices in screening, needs assessment, and resource linkage. During the training, CHWs observed the project director administering the screening/needs-assessment tool with PSH residents. CHWs were then observed by the project director administering the screening/needs-assessment tool to ensure fidelity. As part of the human-centered design process, ongoing CHW supervision was provided weekly at the service-delivery site attended by the entire project team (CHWs, dietitian, program managers, primary care physician from the community-based partner directing outcomes’ management, and university-based faculty). To understand the barriers to food security and safe/healthy cooking and eating practices, the project team reviewed PSH-resident feedback on nutrition activities, the number of people participating in weekly activities, and CHW challenges engaging the population during weekly supervision meetings and made program adjustments as needed to better meet resident needs. Ongoing training took place during the third and fourth quarters for the CHWs and the project dietitian through an interprofessional practice course on social determinants of health at UTHealth Houston McGovern Medical School. This course culminated in a healthy luncheon at one of the two PSH buildings for the project staff/faculty, medical students, and PSH residents to eat lunch together to build community and trust and to reduce implicit bias. 

### 2.6. Measures

CHWs assessed food security using the 10-item U.S. Department of Agriculture (USDA) Adult Food Security Survey which is utilized to collect USDA’s food security statistics and classifies respondents into four categories delineating high, marginal, low, and very low food security, based on a recall period of the past 12 months [36]. This widely used survey instrument was chosen due to its three-stage design with screeners which keeps respondent burden to the minimum and allows for comparison of the survey findings with existing food security statistics [36]. Responses were dichotomized as (1) high or marginal and (2) low or very low food security for the analyses, as per previous research [13,37,38]. While some research supports the USDA survey as a valid and robust measure of food security, other research has also found some evidence that interpretation of survey questions differed among different demographic groups [39]. Accordingly, survey administration was piloted and CHWs were directly observed administering the survey to PSH residents to ensure respondents understood the questions they were being asked, and additional training in survey administration was provided by the project director to improve administration. 

During weekly intervention planning meetings, the project team identified additional questions from the research that were added to the baseline survey as a needs assessment by project members in the following domains to obtain PSH resident information to tailor project activities to improve engagement and impact: (1) resident demographics; (2) fruits and vegetables, consumption habits and perceived barriers to fruits and vegetables; (3) self-reported cooking-skill level; (4) effectiveness of vision for meal planning, grocery shopping, and safe cooking; (5) resident interest in planned project activities; (6) salient food and resource needs to reduce food insecurity; and (7) findings from apartment inspections to assess safe cooking and healthy eating and ongoing needs. The construct validity of the needs-assessment items related to consumption habits and perceived barriers has been established in the research [40] as well as self-efficacy in cooking [41]. However, a whole-grains question was added that was not included in the validated set, and the validity for the overall-cooking-ability-survey items is in the process of being established by the Nourish Program [35]. The effectiveness of vision for meal planning, grocery shopping, and safe cooking were added by a medical student interning with the project as a student-initiated research project. When administering the survey, CHWs were trained to inform participants that the vision items were intended to identify residents who were experiencing vision-related barriers and to refer them to the necessary health services. All data were collected by CHWs using Research Electronic Data Capture (REDCap), a secure, web-based application designed to support data capture for research studies housed at UTHealth Houston [42]. 

### 2.7. PSH-Resident Feedback

As part of the human-centered design process, the project aimed to obtain systematic feedback from PSH residents utilizing an end-to-end approach [26,31,32]. In the initial needs assessment, CHWs obtained systematic feedback from 83 PSH residents on their nutritional needs, preferences, values, and barriers to food security and healthy cooking/eating. Prior to implementing nutrition education sessions, PSH residents were asked to submit nutrition-based questions on comment cards that they wanted the public health dietitian to answer. Following each healthy cooking class, participants were asked by CHWs to provide written feedback on comment cards for the following questions:My Nutrition Goal(s) for Safe and Healthy Cooking/Eating:What did you like the most?What are some new things you learned from this group activity?Suggestions: How can we improve for our next class?What dish would you like to learn to cook next time?

Additional feedback was obtained from 15 participants during the distribution of cookbooks at the end of project to inform ongoing program development and included the following points:What nutrition project activities did you participate in?What nutrition project activities did you enjoy?What did you learn from participating in the project?How was the nutrition program helpful?Describe how you are using what you learned in the nutrition program?

### 2.8. Data Analysis

Food security status was calculated on baseline and follow-up USDA Adult Food Security surveys as follows: Raw score zero—high food security among adults; raw score 1–2—marginal food security among adults; raw score 3–5—low food security among adults; raw score 6–10—very low food security among adults (Table 1). Survey responses were dichotomized as (1) high or marginal and (2) low or very low food security for the analyses, as per the previous research [13,37,38]. Descriptive statistics were used to examine the demographic characteristics (age, gender, race/ethnicity), pre/post food security scores and needs-assessment items. The project team (project leads, dietitian, and lead CHW) used qualitative analysis methods to analyze PSH-resident feedback to guide program planning and implementation. Feedback obtained on comment cards (distributed at group activities) and collected during the needs assessment and food/kitchenware/cookbook distributions was copied into a project spreadsheet stored in a university-based secure drive, according to the corresponding project activity, date, and time. Quality checks on the collection and storage of resident feedback were conducted by the project director and lead community health worker. The aggregated text from the feedback was then cleaned and combined into one document and analyzed using thematic analysis [43,44,45] in which project team members coded the feedback using systematically applied indexing categories [44]. 

Thematic analysis was conducted as an iterative process consisting of six steps: (1) becoming familiar with the data, (2) generating codes, (3) generating themes, (4) reviewing themes, (5) defining and naming themes, and (6) locating exemplars [43,44]. PSH-resident feedback was qualitatively analyzed both during weekly team meetings by all project staff for project improvement and at the end of the project by the project director for grant reporting. Major themes from residents emerging across project activities are presented in Table 2.

## 3. Results

### 3.1. Sample Characteristics

A total of 83 PSH residents completed the baseline USDA Food Security Survey [13,33,34] which included additional needs-assessment items. Age was obtained for 81 residents, ranging from 25 to 73 years (mean age of 55.8 years). The majority of baseline respondents identified as male (*n* = 64; 77.1%), while 18 identified as female (21.7), and 1 identified as a transgender female (1.2%). The majority also identified as Black/African American (*n* = 61; 73.5%), while 13 (15.7%) identified as Non-Hispanic White, 2 (2.4%) identified as Asian, 2 (2.4%) identified as Hispanic/LantinX, 1 (1.2%) identified as American Indian/Alaskan Native, and 1 (1.2%) identified as Arab/Middle Eastern American. 

### 3.2. Baseline Screening Results

As can be seen in Table 1, 53 PSH residents (64%) were found to have low or very low food security during the baseline food security screening with another 10 (12%) having marginal food security. The needs-assessment data found that all 83 PSH residents lacked necessary kitchenware items in their SROs for safe cooking and eating including pots, pans, cooking utensils, cups, dishes, flatware, and small appliances. Seventy-seven participants answered the vision-related questions in the needs assessment which helped CHWs to identify barriers to purchasing and preparing healthy food. Thirteen residents (16.8%) reported that their ability to read nutrition labels was poor. Only seven (9%) reported that their eyesight was poor when buying food, and only two (2.6%) reported their eyesight was poor when preparing meals. Persons reporting poor vision were connected with PSH case managers to follow-up for a medical evaluation. 

### 3.3. Follow-Up Surveys

CHWs attempted to administer a follow-up food security survey 90 days after baseline; however, several attempts often had to be made to engage some residents and collect survey data. Accordingly, the mean number of days between baseline and follow-up surveys for the 60 PSH residents who completed both a baseline and follow-up was 135 days (standard deviation: 49.64 days). A slight majority, 33 residents (55%), had improved food security scores or were ascertained as having high food security (zero score). However, food insecurity remained a problem for many PSH residents possibly due to the cut of SNAP emergency allotments (EAs)—temporary benefit increases that Congress enacted to address rising food insecurity and provide economic stimulus during the COVID-19 pandemic—that ended after February 2023 issuances. In Texas, every Texan household began receiving at least $95 a month less in SNAP benefits (with some households seeing reductions of $250 a month or more). 

### 3.4. Individual Level CHW Intervention

The first PSH building was in close proximity (1 min walk; 393.7 ft) to the faith-based partner organization hosting the bi-weekly food distribution; however, the second PSH building required a 12 min, 0.5-mile, walk. Hence, an initial activity conducted by the CHWs was to identify residents with mobility impairments and transportation barriers to attending the bi-weekly food distribution. To improve access to food distribution items that included fresh fruits and vegetables and pantry staples (e.g., canned foods, dried beans, pasta and rice), CHWs worked with PSH staff to coordinate the delivery of boxed items from bi-weekly food distribution events to approximately 72 of the PSH residents with mobility impairments as recommended in the prior research [13]. CHWs also ensured all eligible participants were linked with SNAP and WIC benefits. 

The lack of necessary kitchenware was addressed through the creation of an Amazon Wish List by leadership at the faith-based partner organization in order for church members to purchase kitchenware items on the wish list to be distributed by the CHWs. As the project progressed, grant money was repurposed to purchase the necessary kitchenware not fulfilled through the wish list for all 83 residents completing the needs assessment. CHWs received feedback from residents with dental issues who reported benefiting from receiving blenders to make their food easier to eat. Cooking safety was increased by providing air fryers for residents, many of whom were reported as having safety issues with stove-top cooking. 

One-on-one resident support provided by CHWs in resident SROs was fully implemented by the second quarter. A total of 294 separate individual-level resident interventions were conducted by the end of the 1-year pilot with 140 PSH residents. The most frequent type of activity completed by the CHWs (*n* = 228 sessions) included pantry inspections for food insecurity/nutritional deficiency in resident SROs and/or follow-up on referrals received from PSH staff to intervene with individual residents experiencing food insecurity. In their SROs, CHWs engaged food insecure PSH residents in meal planning, label reading, and delivery of healthy foods through (1) the distribution of 176 large boxes of food (fresh vegetables and nonperishable items from the bi-weekly distribution provided by the regional food bank); (2) linkage to food pantries (both through the distribution of flyers with directions and/or accompanying residents to pick up food; (3) assisting residents with SNAP benefits in signing up for Amazon Fresh deliveries since there was not a close-by grocery store; and (4) providing residents with necessary kitchen items so they could cook and eat healthy food in their SROs (pots/pans, dishes, utensils/flatware, and small cooking appliances). 

As recommended by Bowen et al., 2019, linking residents with food delivery was a particularly successful strategy [13] as indicated by the following resident feedback. “She [CHW] got me involved with Amazon Fresh—I miss her… it is helping out a lot—I got the refrigerator filled... I’m cooking off and on—still don’t have taste or smell because of COVID and I need to go to the doctor… I can’t cook like I want to—but my freezer is full”. 

The second most common individual-level activity provided by CHWs was social services assistance that included assisting residents with SNAP/WIC applications and linkage to primary care (*n* = 19 sessions). Some residents were distrustful of existing co-located social services staff and formed bonds with the nutrition-focused CHWs who were able to assist residents in obtaining needed benefits to close the SNAP coverage gap. CHWs provided wellness checks and emergency food assistance for eight residents who were not leaving their SROs and were believed to not have food. 

One-on-one nutrition education provided by the public health dietitian was provided in SROs for five residents. Three residents received one-on-one cooking supervision and food safety training in their SROs provided by CHWs. One-on-one CHW support was also provided for a resident with social anxiety during a trip to the grocery store and for a non-English-speaking resident with very low food security who was connected with culturally affirming healthy food items. Other resident support provided by CHWs included transportation to medical appointments (six sessions total) for two aging and disabled residents with chronic illnesses, including one resident who had advanced cancer. 

### 3.5. Group-Level CHW Intervention

Ten nutrition education sessions were implemented by the public health dietitian during the third quarter of the project. As part of the planning process, the dietitian met with the CHWs weekly to plan nutrition education sessions that addressed the nutrition-based questions from the PSH residents. In particular, the residents asked for information on preparing meals to support the management of diabetes and heart disease and weight loss, having a balanced diet using meat-alternatives, and the importance of purchasing organic foods. In addition to addressing resident questions, session topics planned by the dietitian included nutrition-label reading, food safety (cooking temperatures/use of thermometers, storing food, defrosting meat, and the meaning of sell-by dates), and knife skills. The CHWs and dietitian also engaged the residents in playing a series of weekly nutrition games that the dietitian developed for this project including The Price is Right; MyPlate Nutrition Bingo; Nutrition Crossword Puzzles; Nutrition Pictionary; and Nutrition Jeopardy. The nutrition games were very popular among the residents, and multiple comments were obtained on the new knowledge they had gained to improve healthy eating. In particular, residents reported learning how to purchase and prepare healthier food such as low-salt canned goods, corn tortillas, natural peanut butter, and Greek yogurt. One resident commented “I enjoyed the nutrition sessions the most… I was homeless for two years and needed to be reminded of the basics again like how to slice avocados and tomatoes… practicing during the nutrition classes was helpful”. Several residents also commented on the positive experiences they were having in a group activity with other residents. Two interviewees reported learning to spell and pronounce vegetables correctly. When asked what nutrition program activities he enjoyed, one resident described the nutrition games as “a lot of enthusiasm around it—I learned how to spell squash, turnips, collard greens—different vegetables… fruit juices… was a lot of fun as well… loved the program… loved the games—the program was very good”. During a nutrition game, a resident responded enthusiastically “You can add yogurt to chili? I didn’t know that!” Another resident reported “I am using a lot of vegetables and fruits now… I cook apple soup and cook vegetables for my noodles”. Another interviewee reported that the games helped him to “learn about the different fruits and vegetables and the right amount of salt”. A resident reported that his favorite activity was the nutrition games because “during the nutrition games… instead of everybody hating on each other, we came together as one as we should be… enjoyed all that”. Another interviewee described how he met other residents through the nutrition games reporting “I was in the hospital… I have kidney and liver problems... I met some good and courteous people [during the nutrition games]… I am from Louisiana and I know how to cook… but I learned about healthy seasoning”. Another resident reported that she benefited from learning “price points and nutritional information about the food”. A resident with diabetes reported that during the nutrition games “I won… was pretty good in the games… I am a diabetic and knew stuff from diabetic classes and won games… knew the answers from Dawn Center [Diabetes Awareness and Wellness Network—diabetes self-management education program in the region]… I graduated… they [DAWN] teach you all the things you need to know—about grains and fibers—when they [dietitian and CHWs with the project] were asking questions I was popping up—I got ahead of a whole lot of people”. 

Twenty healthy cooking classes were conducted from the second through to the fourth quarters, and class participation ranged from 2 to 16 participants, with a mean number of participants 7 per class (138 participants total). The kitchen equipment utilized in the cooking class was modeled after the kitchens in resident SROs (two burner stoves and a microwave oven), and the residents participated in a community meal after each class. Food items including fresh fruits and vegetables and pantry staples from the bi-weekly food distribution (donated by the regional food bank)) were used to teach residents how to cook healthy meals with foods they could access through the food bank. A major theme arising from the healthy cooking classes was related to expanded preferences for and attitudes towards healthy foods residents could easily prepare in their SROs. Resident feedback provided on comment cards after each class identified popular recipes including black bean soup, water infused with cucumbers, potato scramble, easy bean chili with fresh tomatoes and mustard greens, apple crisp in a mug using oatmeal packets from the food distribution to make the crispy topping, and omelets in a mug with eggs, green and red bell peppers, spinach, shredded Italian or cheddar cheese, milk, and salsa (if desired). During a weekly staff meeting, resident feedback was reviewed and a CHW reported that “the residents were initially put off and uninterested in cutting up and making their own omelets, but they came around to it later… many of them were also taken aback to cook the eggs in the microwave, rather than the stove, and were unsure how they would taste, however, all of the residents who made the eggs enjoyed it”. 

Additional class favorites included yogurt parfaits with Greek yogurt, granola, raspberries, blueberries, and honey. One CHW reported during a planning meeting that “residents seem to really enjoy yogurt and being able to pick out the toppings”. After participating in the fruit and vegetable smoothie stations, some residents expressed wanting a blender to make their own smoothies in order to consume more fruits and vegetables. Feedback obtained following the healthy cooking class, taco stations, indicated residents enjoyed cutting their own vegetables and making spice blends, which were added to black beans. One resident reported that during a cooking class “I learned things about herbs I did not know”. A theme emerged from the analysis of the feedback on the comment cards that regaining food preparation practice skills lost during homelessness such as using knives to cut vegetables during cooking classes was a particularly important learning activity (mentioned by seven residents on comment cards). One resident reported that “learning about spices in the classes brought it back to me… how to cook… once you get it… taste buds help to create your own spice mix”. Another resident reported “All of it [cooking classes]... especially the black bean soup… I like to cook—to learn… I had forgotten a little... and I learned a whole lot… I really did need it… I am now cooking Spanish rice and beans and making salads”. An additional theme that emerged was that participating in the cooking classes exposed them to healthier food options that they began using when cooking on their SROs. One resident reported during a door-to-door distribution event “I did everything [in the cooking class]… cut the onions… used cold water… I prepped and cooked chili.. I learned to cook chili a different way… I put chopped onions in the air fryer I got from the program”. 

Two grocery store pop-ups and six grocery store field trips were conducted by CHWs to engage residents in learning how to shop for healthy foods. Approximately 30 PSH residents attended each pop-up event to access donated foods. The number of PSH residents attending grocery store field trips ranged from three to six per trip (mean number of five participants per trip). Project team members met residents (who were transported via Lyft) at a grocery store (a 7 min and 3.2-mile drive from the PSH buildings), which most residents had not shopped at before due to transportation barriers. As part of the field trip, the dietitian and CHWs engaged participating residents in learning how to shop for healthy and affordable food items. During each field trip, participants were given donated gift cards to purchase an item from each food group. CHWs also purchased insulated bags for their groceries and glass containers to store leftovers in. A major theme that emerged related to the grocery store fields trip was related to community re-entry and learning to grocery shop for healthy foods as a group. One resident commented that “the best part of being here is being with my friends”. Another resident commented that she was not purchasing items yet and reported “I am walking around learning and taking notes about what is here”. Another resident was excited to learn about all the different spices that were available that could be used in place of salt as he reported having hypertension. Another resident reported benefitting from the two grocery store field trips he participated in reporting “I hadn’t ridden in a car in a long time… being able to go to the grocery store with others was helpful because this is not something I would do by myself”.

Three community meals were served during special occasions and were well attended including Thanksgiving, which had 16 PSH residents attending, the medical student luncheon with approximately 25 PSH residents attending along with medical students and project staff, the mental health wellness luncheon which had approximately 25 PSH residents attending, and the program graduation which had 28 PSH residents attending. One resident commented that “I enjoyed the community meals because I like to visit with others… and when I go to the grocery store, I always buy the same thing… during the lunches I tried a variety of food… it was good food”. 

Four community gardening events were conducted that included 11 different PSH residents (3–4 participants per event). Prior to beginning the gardening component of the project, leadership toured a large farm run by one of the pastors at the faith-based community partner. Additionally, CHWs toured a university-based teaching garden run by the public health dietitian at the school of public health. CHWs obtained a small grant from a local native plants nursery, which was used to purchase seeds and herbs. Using grant funds and in-kind contribution from the school of public health, grow bags and soil were purchased for residents. Gardening activities included planting herbs in planters at one PSH building and planting herbs and vegetables in individual grow bags at the other PSH building. To encourage autonomy, each resident received their own grow bag and chose which vegetables/herbs to plant (lettuce, collard greens, kale, etc.). A major theme that emerged was resident ownership of maintenance of the community garden. One resident referred to the plants in her garden grow bag as “her babies” and formed a watering group with another resident to ensure the plants in the grow bags received enough water. Residents in the gardening group also identified times they could go to the garden together outside of the gardening classes, which helped to build community among the residents. Another resident reported during the key informant interviews that community gardening was one of his favorite program activities because “when you see it green and growing it makes me feel good”. 

A resident cookbook containing popular and easy-to-prepare recipes from the healthy cooking classes was developed, printed, and distributed to PSH residents who participated in the project. Recipes were separated by cooking equipment needed (microwave, stovetop, cutting board) and type of recipe (dressings and sauces, spice blends). The cookbook included pictures of residents from different cooking classes preparing the meals. The resident cookbook can be found on the website (https://sph.uth.edu/research/centers/dell/nourish/research-resources/Temenos_Cookbook.pdf; accessed on 7 January 2024).

## 4. Discussion

Our finding that 64% of PSH residents screened by CHWs at project intake were experiencing low or very low food security is similar to the rate of low or very low food security found among PSH residents aged 45 years and older in Los Angeles, California (67%) [13]. Implementation studies for reducing food insecurity and increasing nutritional outcomes among PSH residents have received very little attention in the literature to date [18]. Thus, this project both extends the generalizability of previous findings and provides preliminary food security outcomes and resident perceptions following the implementation of evidence-based nutrition strategies. The human-centered design enabled the project planners to keep the PSH residents at the center of project development and implementation and to tailor strategies as needed to meet the residents where they were at. The analysis of comments collected after project activities and during the distribution identified themes of needing additional equipment and cooking-skills practice to safely prepare healthy food in their SROs. Research on food security among the formerly homeless youth in PSH suggests that while housing removes some major sources of food insecurity from their lives, it adds others [19]. In line with our study, Brothers et al. (2019) found similar individual mechanisms associated with food insecurity in PSH including limited cooking skills, equipment, and coping strategies [19]. Similar structural mechanisms were also found including the location of the PSH building in a food desert [19]. Hence, the low level of food security identified across multiple PSH programs points to a need for the implementation of effective approaches [13,19]. A primary finding from the initial needs assessment was that all 83 PSH residents lacked adequate kitchenware items in their SROs for safe cooking and eating including pots, pans, cooking utensils, cups, dishes, flatware, and small appliances. This finding has been linked in prior research pointing to a relationship between lacking adequate kitchenware items and consuming more unhealthy foods [46]. Thus, tailoring project implementation to address this need was an important human-centered design strategy. 

Systematic feedback obtained from the residents indicates that the group activities, particularly the nutrition games developed by the public health dietitian for the project, were well received and perceived as beneficial for helping residents to improve access to nutrition benefits, facilitate the acquisition and application of nutritional knowledge and cooking skills, and to develop community with other PSH residents. Additionally, comments obtained following the group cooking classes indicate that residents expanded their preferences for and attitudes towards healthy foods they could easily prepare in their SROs. This finding is aligned with research findings that cooking components of nutritional programs are an effective strategy compared to nutrition education (knowledge-, attitude-, and awareness-centered approaches) alone in changing diet [47]. Comments made by PSH residents indicate that, in addition to increased knowledge, an improvement in self-efficacy was also attained, which has been found to be an important component in improving healthy meal preparation after participation in nutrition education [48]. This is a particularly salient finding as the analysis of the resident comments also revealed that participation in the group activities was perceived as particularly helpful for regaining both cooking skills and feelings of social connectedness that were lost during periods of chronic homelessness.

### Limitations

Our findings should be considered in light of the study’s limitations, including the implementation of the project at a single PSH program which limits generalizability. Additionally, our study was subject to selection bias as not all 140 PSH residents participated in the project. While 83 PSH residents completed informed consent and the baseline survey/needs assessment, the uneven engagement of residents may have biased the feedback on project activities. Additionally, the survey and needs-assessment data were obtained through self-report and were subject to self-report bias. While the project aimed to continuously utilize best practices in the implementation of data-driven evidence-based practices, constraints of engaging residents within the low-resource PSH environment contributed to additional limitations such as variation in the time between the baseline and follow-up surveys. Resource limitations presented additional constraints on the ability of staff to implement more standardized measure such as a self-efficacy scale, which would be a recommendation for future research. A final limitation included the underestimation of food access barriers during the initial implementation of the project. The project initially experienced challenges to having adequate food for cooking classes, community meals, and emergency food distribution. To overcome this challenge all project members worked on leveraging partnerships through the faith-based partner organization including a local farm and the regional food bank, re-allocated grant funds, leveraged funds from other projects, and increased new funding sources through small-grant applications to local organizations and foundations. 

## 5. Conclusions

This pilot project describes the innovative use of a community benefits grant to reduce food insecurity and improve nutritional outcomes through a multi-level project for formerly chronically homeless adults in PSH using a human-centered design methodology. The high level of food insecurity identified through baseline screening in this project is similar to the level of food insecurity found in PSH in another area within the U.S., providing additional evidence of the need to address food insecurity along with housing to improve outcomes for homeless populations. All PSH residents were targeted for participation, and a high level of engagement among a core group of PSH residents in a range of nutrition-focused CHW interventions continued throughout the project. Accordingly, the success of this project points to a need for additional research and demonstration projects in this area to address this important public health problem. 

## Figures and Tables

**Figure 1 ijerph-21-00108-f001:**
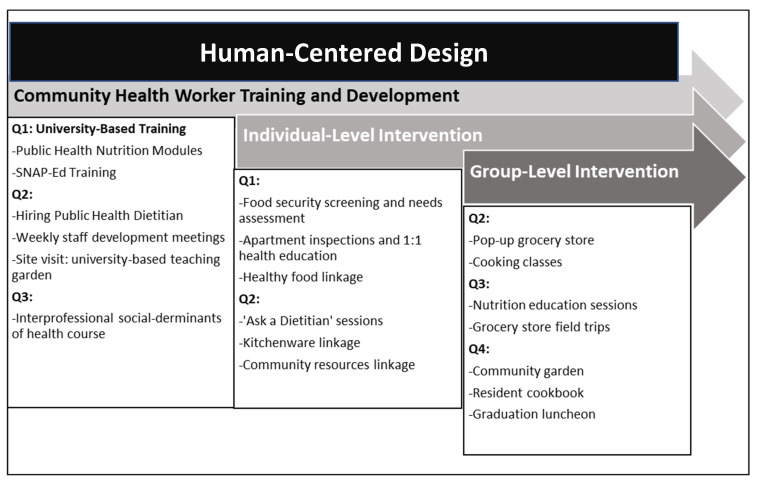
Project-implementation timeline (Quarters 1–4).

**Figure 2 ijerph-21-00108-f002:**
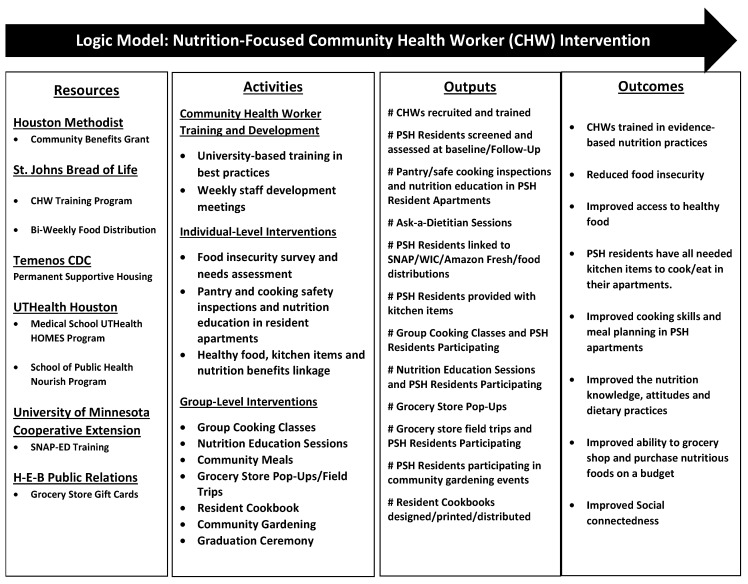
Logic model.

**Table 1 ijerph-21-00108-t001:** PSH food security screening results.

Raw Score	Status	Baseline Numbers/Percentages (*n* = 83 Residents Screened)	Follow-Up Numbers/Percentages (*n* = 60 Residents Screened)
Zero	High Food Security	20 (24%)	16 (27%)
1–2	Marginal Food Security	10 (12%)	14 (23%)
3–5	Low Food Security	24 (29%)	13 (22%)
6–10	Very Low Food Security	29 (35%)	17 (28%)
3–10	Low/Very Low Food Security	53 (64%)	30 (50%)

**Table 2 ijerph-21-00108-t002:** Major themes from residents across project activities.

Data Source	Project Activities	Major Theme(s)
Needs Assessment in REDCap	Baseline Screening	Lack of necessary kitchenware items in resident SROs for safe cooking and eating
Nutrition-based questions requested from residents	Nutrition Education Sessions	Knowledge gaps on how to purchase and prepare healthier food
Comment Cards	Nutrition Education Sessions	Positive perceptions of healthy food options
Comment Cards	Group Cooking Classes	Expanded preferences for healthy, easy-to-prepare foods
Comment Cards	Grocery Store Field Trips	Community re-entry
Comment Cards	Community Garden	Resident ownership
Brief feedback: enjoyment, learning, helpfulness, and application of nutrition activities	Cookbook Distribution	Regaining cooking skills after homelessness Positive experiences with other residents

## Data Availability

The deidentified data presented in this study are available on request from the corresponding author.

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
