# Peer review of "Pilot Implementation of a Nutrition-Focused Community-Health-Worker Intervention among Formerly Chronically Homeless Adults in Permanent Supportive Housing"

_ijerph, 2024, doi:10.3390/ijerph21010108_

Round 1

Reviewer 1 Report

Comments and Suggestions for Authors

Introduction

§  Paragraph 2, last sentence is an incomplete sentence. It seems as though some information may be missing in the last part of the sentence, or it is meant to explain the preceding existing data on FI rates among the PSH population.

§  The “I” in innovative of the start of the 3rd paragraph needs to be capitalized.

§  The overall purpose of the study needs to be better identified and clarified in this section.

Methods

§  The “human-centered design methodology” is referenced several times throughout the paper, yet it is not adequately described and referenced.

§  Logic model and CHW Training and Development figures are included, but could be condensed to provide information that is more relevant to the project activities and timeline.

§  This section also goes back and forth between describing activities/measures for the CHWs and for the adults in PSH, which is quite confusing.

§  The measures used in this study need to be better defined and clarified, including a timeline for when each measure was taken. It appears that a needs assessment, qualitative data, key informant interviews, and direct observations took place. This needs to be distinguished better in this section for clarification. This would allow for a better understanding of the Discussion section as well.

§  Food security as measured by the USDA survey has several versions (e.g., 18-item, 10-item, 6-item), yet it was not described which version was used for this study.

§  More information of how the needs assessment instrument was developed, including any reliability/validity information, should be included.

§  Were any interview guides developed as part of the qualitative data collection? If yes, this also needs to be described

§  Data analysis needs to be further described as to how the quantitative and qualitative data were analyzed. For the qualitative data, did audio/video recordings take place, and were these files transcribed verbatim and cleaned prior to analysis? There needs to be much more information for how especially the qualitative data was collected, analyzed, and interpreted.

Results

§  This section is rather long and disjointed, and varies between presenting results from CHW feedback and PSH participant feedback.

§  There is a lot of information in this section that does not need to be reported. For example, there is quite a lot of text about some of the food items, what they included, why those items were included, and why they were made in that way. This takes up a lot of space and detracts from what the intended purpose of the study is.

§  Including tables of information that display relevant quantitative and qualitative information would have been helpful in understanding noteworthy and/or significant results.

§  It was not clear when the pre-post surveys were administered.

§  This section also includes information regarding three vision questions that were included in the Vision section of the needs assessment. For the first question, “How would you rate your ability to read nutrition labels” it seems that unless specifically asked regarding vision, that participants could also take this to mean understanding HOW to read a food label. Also, this information should be included in the measures part of the Methods section, and result should be specifically reported here.

§  There is some redundance in text that needs to be cleaned up for clarity.

§  It appears much of this text could be pared down and described better in the Discussion section.

Discussion

§  A Limitations subsection is needed in this area.

§  This section was rather short and did not relate the relevance/noteworthiness of this study to other previous studies, or how it informs the current literature base.

Overall

§  More attention to detail in grammar and sentence structure. While the article overall addresses a much needed gap, some of the sentences were rather long and/or left out punctuation that would have greatly improved readability. Please make sure to do a thorough read-through of the manuscript to clean up and address language and sentence structure.

§  I would also recommend that this study be split into two separate studies. The training, planning, and preparation of the CHWs could be considered one separate article, while the results of the intervention could be another study. This might add further clarity to all of the data that was reported in this article.

Comments on the Quality of English Language

English language is fine. However, grammar, punctuation, and sentence structure need to be greatly improved for overall better readability.

Reviewer 2 Report

Comments and Suggestions for Authors

The article is a valuable one and I congratulate the authors for it. It makes an important contribution to the field of food security, but in order to increase the credibility of the research, certain clarifications are imperatively necessary, which are not mentioned or not sufficiently explicit, as the case may be. Without their operation, the article cannot claim that the presented research is eloquent, although it is a very well documented material.

I recommend the authors to better clarify the following important aspects:

1. What were the criteria for selecting the sample?

2. How was the questionnaire validated? Who validated it?

3. What were the exclusions (if any) from the questionnaire and respondents?

Later, these aspects must be included both in the abstract and in the conclusions.

Round 2

Reviewer 2 Report

Comments and Suggestions for Authors

All my comments were resolved positively and the manuscript was greatly improved.